# Diseases, Health-Related Problems, and the Incidence of Malnutrition in Long-Term Care Facilities

**DOI:** 10.3390/ijerph20043170

**Published:** 2023-02-10

**Authors:** Jos W. Borkent, Hein P. J. Van Hout, Edith J. M. Feskens, Elke Naumann, Marian A. E. de van der Schueren

**Affiliations:** 1Department of Nutrition and Health, HAN University of Applied Sciences, Kapittelweg 33, 6525 EN Nijmegen, The Netherlands; 2Division of Human Nutrition and Health, Wageningen University, Stippeneng 4, 6708 WE Wageningen, The Netherlands; 3Amsterdam University Medical Center, Department of General Practice and Medicine for Older Persons, Vrije Universiteit, Van der Boechorsstraat 7, 1081 BT Amsterdam, The Netherlands

**Keywords:** nursing homes, undernutrition, older adults, nutritional status, InterRAI, long-term care facilities

## Abstract

Certain diseases and malnutrition are known to co-occur in residents of long-term care facilities (LTCF). We assessed which diseases and health-related problems are associated with malnutrition at admission or with incident malnutrition during stays and how different definitions of malnutrition affect these associations. Data of Dutch LTCF residents were obtained from the InterRAI-LTCF instrument (2005–2020). We analyzed the association of diseases (diabetes, cancer, pressure ulcers, neurological, musculoskeletal, psychiatric, cardiac, infectious and pulmonary diseases) and health-related problems (aspiration, fever, peripheral edema, aphasia, pain, supervised/assisted eating, balance, psychiatric, GI tract, sleep, dental and locomotion problems) with malnutrition (recent weight loss (WL), low age-specific BMI (BMI), and ESPEN 2015 definition (ESPEN)) at admission (n = 3713), as well as with incident malnutrition during stay (n = 3836, median follow-up ~1 year). Malnutrition prevalence at admission ranged from 8.8% (WL) to 27.4% (BMI); incident malnutrition during stay ranged from 8.9% (ESPEN) to 13.8% (WL). At admission, most diseases (except cardiometabolic diseases) and health-related problems were associated with higher prevalence of malnutrition based on either criterion, but strongest with WL. This was also seen in the prospective analysis, but relationships were less strong compared to the cross-sectional analysis. A considerable number of diseases and health-related problems are associated with an increased prevalence of malnutrition at admission and incident malnutrition during stays in LTCFs. At admission, low BMI is a good indicator of malnutrition; during stays, we advise use of WL.

## 1. Introduction

With the ageing population in Europe, the share of people aged >65 years will increase to 30% by 2055 [1]. This process will lead to an increased number of frail, older adults who are highly care-dependent [2]. Dutch government policy aims to allow older adults to live at home for as long as possible. However, when care requirements become too high and care can no longer be provided at home, admission to a long-term care facility (LTCF) may be inevitable [3]. As a consequence, residents in LTCFs often suffer from multiple medical conditions, such as dementia, neurological diseases, diabetes, and cardiovascular diseases [4].

The high disease burden of residents in LTCFs increases their risk of becoming malnourished. In older adults, malnutrition mainly refers to protein–energy malnutrition, which leads to loss of weight and eventually to a low BMI [5]. Even when BMI does not drop below clinically relevant cut-off values—for instance, in those with obesity [6]—involuntary weight loss in older adults is always a risk [7]. Disease activity in older adults can reduce residents’ ability to consume or digest food [8,9,10,11], while inflammation can increase protein and energy needs. Worldwide, about 17.5–28.7% of all residents in LTCFs are reported to be malnourished [12,13]. Malnutrition, in turn, can increase the risk of developing diseases, thus creating a perpetual circle where diseases and malnutrition strengthen each other [14]. 

As admission to a LTCF marks the latest phase in life, residential care should focus on function preservation and optimizing quality of life. A good nutritional status can contribute to this, as malnutrition is associated with function loss and impaired quality of life [15,16,17]. To provide optimal nutritional care directly after admission and during stay, it is important to know which diseases are related to being or becoming malnourished. Previous studies have shown that several diseases (e.g., dementia, depression, cancer, pressure ulcers and COPD) and health-related problems (dysphagia, eating and chewing problems) are associated with malnutrition in LTCF residents [18,19]. However, most studies conducted so far used cross-sectional data, collected at a non-specific timepoint during stays, providing no evidence about time of onset (before or after admission) of malnutrition. It is therefore unclear whether residents with diseases and health-related problems were already malnourished at admission, or became malnourished during their stay. 

In addition, previous studies used different criteria to assess malnutrition (recent weight loss, low BMI, or the combination of both (ESPEN 2015 criteria [5])), which could have affected the results. For example, residents with cardiometabolic diseases, which are related to a high BMI, are not likely to trigger the cut-off value for low BMI, despite recent weight loss. On the other hand, in slowly developing and chronic diseases like chronic pulmonary disease and neurological problems, weight loss may also occur slowly and, therefore, the criteria for malnutrition based on >5% of >10% reduction in body weight over 1 or 6 months, respectively, will not be met. In addition, the reported prevalence rates are expected to represent an underestimation of malnutrition in LTCFs. At the same time, it remains unclear how different criteria to define malnutrition affect prevalence rates and which definitions can be used best to identify participants at risk of malnutrition at admission and during a LTCF stay. 

In this study, we assessed the prevalence of diseases and health-related problems in LTCF residents at admission and during stay, and investigated their relation to recent weight loss, low BMI, or a combination of both. 

## 2. Methods

### 2.1. Data Source

For this research, we used cross-sectional, as well as prospective, data obtained from the Dutch InterRAI database for Long-Term Care Facilities (InterRAI LTCF). These data have been previously described by our group [20]. 

InterRAI LTCF is an instrument for healthcare providers to systematically obtain information of about health conditions of LTCF residents. InterRAI LTCF is a so-called minimum dataset (MDS), first implemented in the Netherlands in 2005, covering up to ~50 facilities in 2020. Trained nurses use a standardized assessment form to assess residents’ health status. The assessment form also contains questions regarding nutritional status (weight loss and BMI) and presence of diseases. 

Each new resident is screened within a month after admission on average (admission assessment). Thereafter, follow-up assessments (same assessment form) are performed every 3–6 months. When a new LTCF first starts to use InterRAI, all existing residents are screened, despite not being recently admitted. This measurement can be seen as a “delayed first assessment” in InterRAI. 

Dutch InterRAI subjects (aged ≥ 65 years) living in LTCFs between 2005 and 2020 were included in this study. Two groups were formed: the “newly admitted” (those with an admission assessment) and the “existing” residents (those with a “delayed first assessment”). So, the difference between these two groups is the time elapsed between their admission and first assessment, which was shorter in the “newly admitted” group and longer in the “existing” residents group. 

In total, 4190 residents with an admission assessment were available; median time to first assessment was 16 days [IQR: 7–30] after admission. In total, 5592 residents with a delayed first assessment were available, and this assessment took place with a median time of 345 [IQR: 117–914] days in male and 546 [IQR: 165–1363] days in female residents after their initial admission. 

### 2.2. Inclusion Criteria

Data of admission assessments (“newly admitted” residents) were used to provide an overview of characteristics of the residents at admission and to perform cross-sectional analyses. Inclusion criteria were: age ≥ 65 years and available data on BMI and weight loss. Exclusion criterion was: presence of end-stage disease, i.e., terminally ill residents with a life expectancy < 6 months as indicated by the treating physician, to exclude residents with incurable malnutrition. The total number of included residents for cross-sectional analysis was 3713.

Prospective analyses were performed for residents (both “newly admitted” and “existing”) with more than one measurement available. In addition to the inclusion criteria described above, participants were excluded when they were malnourished at first available measurement (defined as having low age-specific BMI < 20 kg/m^2^ for residents < 70 years, or <22 kg/m^2^ for residents ≥ 70 years, or having recent weight loss (5% body mass in the last 30 days or 10% in the last 180 days)), or having end-stage disease at first or/and last available measurement. Total number of included residents for prospective analysis was 3836. Numbers and reasons for exclusion are shown in Figure 1.

### 2.3. Measurements

#### 2.3.1. Diseases

Within InterRAI, only physician-documented diagnoses are registered by checkboxes on the assessment form [21]. The following disease groups are used (with sub diagnoses in parentheses): neurological diseases (Alzheimer’s disease, other type of dementia, hemiplegia, paraplegia, quadriplegia, multiple sclerosis, Parkinson’s disease, stroke/CVA), musculoskeletal diseases (hip or other fracture), cardiac diseases (coronary heart disease, congestive heart failure), psychiatric disorders (anxiety, bipolar disorder, depression, schizophrenia), infectious diseases (pneumonia, urinary tract infection), diabetes, cancer, chronic pulmonary disease, and pressure ulcers. 

#### 2.3.2. Health-Related Problems

Checkboxes are used in InterRAI to indicate the following health-related problems (with sub diagnoses in parentheses): balance problems (falls last month, difficulties self-standing, difficulties turning around, dizziness, unsteady gait), psychiatric problems (abnormal thought process, delusions, hallucinations), GI tract problems (acid reflux, constipation, diarrhea, vomiting), sleep problems (difficulty falling/staying asleep, too much sleep), dental problems (broken teeth, mouth pain, dry mouth, chewing problems, gum inflammation), aphasia, pain, locomotion (independent/with walking device/wheelchair/bedbound), eating help (independent (set-up help only)/supervised), aspiration, fever, peripheral edema. 

#### 2.3.3. Malnutrition

Three different (sets of) criteria for malnutrition were used: recent weight loss, low age-specific BMI and the ESPEN 2015 definition for malnutrition [5]. Recent weight loss (WL) was defined as a loss of 5% body mass in the last 30 days or 10% in the last 180 days. For low age-specific BMI (BMI), a cut-off value of <20 kg/m^2^ was used for residents younger than 70 years and <22 kg/m^2^ was used for residents 70 years or older [5,22]. The ESPEN 2015 criteria (ESPEN) consist of having either low age-specific BMI and weight loss, or having a very low BMI (<18.5 kg/m^2^). 

In InterRAI, data regarding height (at admission) and weight (max 30 days old) were obtained from the patient files. Measurements for both parameters are based on standardized protocols in the LTCF. Previous research showed high reliability of both the weight and length measurements [23]. 

### 2.4. Statistical Analyses

All analyses were performed using SPSS 25. Descriptive statistics (mean with SD, or number with percentage) were used to describe characteristics of residents. Continuous data was checked for normality by using Q–Q plots and stem-and-leaf plots. 

Logistic regression models were used for all cross-sectional analyses. In these analyses on “newly admitted” residents’ diseases and health-related problems, dichotomized as having 0 or ≥1 disease within one disease group were used as independent variables and three criteria for malnutrition (WL, BMI or ESPEN) as separate dependent variables. Gender, age, age, year of admission (2005–2009, 2010–2014, 2015–2020) and living status before admission were added as covariates in the multiple logistic regression models, as these factors are known to be related with malnutrition [12]. 

For the prospective analyses, the time to event (being malnourished) was defined in days as measured from the first available assessment until the first follow-up measurement when a resident was categorized as being malnourished. If a resident was categorized as malnourished, all further follow-up measurements were ignored. If a resident was not malnourished in any follow-up measurement, the resident was censored at their latest measurement. 

Results were visualized by Kaplan–Meier curves to check for the proportional hazard assumption. Thereafter, Cox proportional hazards regression analyses were performed with diseases and health-related problems as independent variables and the three criteria for malnutrition (WL, BMI and ESPEN) as separate dependent variables, adjusting for gender, age, year of admission (2006–2009, 2010–2014, 2015–2020) and living status before admission. 

Post-hoc analysis was performed to test the association between total number of diseases/health-related problems (total numbers were based on previously described categories) and malnutrition.

#### Effect-Modification

A previous study on this data performed by our group showed effect modification by gender and type of first assessment (“admission assessment” vs. “delayed first assessment”) [20]. Therefore, effect modification was tested based on interaction terms between diseases/health-related problems and gender, and type of first assessment. As none of these terms were significant (*p* < 0.05), results were not stratified. 

## 3. Results

### 3.1. Cross-Sectional Analyses at Admission

As shown in Table 1, most residents were female (67.5%), with a mean age of 83.1 years (SD:7.1) and a normal BMI (mean 24.7 kg/m^2^, SD:4.6), and were living alone before admission (64.7%). Most frequently reported diseases and health-related problems were balance problems (70.6%) and neurological diseases (65.4%). On average, residents had 4.6 (SD:2.3) diseases.

Prevalence of malnutrition varied from 8.8% (recent weight loss) to 27.4% (low age-specific BMI). Based on the ESPEN definition, 9.5% of all residents were malnourished. As shown in Figure 2, within the population that was malnourished based on the ESPEN definition, most (69.5%) had a very low BMI (<18.5 kg/m^2^). In total, 31.7% were malnourished based on any definition. 

At admission, nearly all diseases and health-related problems were associated with higher prevalence of malnutrition defined by WL (Table 2). The strongest associations were found for being bedbound (OR: 4.80 [95%CI: 2.90–7.96]), having pressure ulcers (OR: 2.33 [95%CI: 1.68–3.22]) or having fever (OR:2.22 [95%CI: 1.02–4.83]). 

When malnutrition was defined by either low BMI or ESPEN, results were different. Cardiac diseases, diabetes, peripheral edema and walking with a walking device were associated with a lower prevalence of malnutrition. 

Diseases and health-related problems that were strongest related to being malnourished based on ESPEN were being bedbound (OR:2.10 [95%CI: 1.26–3.48]), pressure ulcers (OR:1.81 [95%CI: 1.30–2.53]) and supervised eating (OR: 1.63 [95%CI: 1.28–2.07]). For all associations, comparable results were seen for low BMI and ESPEN, but odds ratios were smaller for low BMI.

Post-hoc analysis showed that an increased number of diseases/health-related problems was associated a with higher prevalence of malnutrition based on WL (OR: 1.21 [95%CI: 1.15–1.27]). This was also seen for ESPEN, although to a lesser extent (1.06 [95%CI: 1.01–1.16]), but not for low BMI. 

### 3.2. Prospective Analyses 

In total, 3836 unique residents had one or more follow-up measurements (Table 3). Total residents’ follow-up time ranged from 5522 (low BMI) to 5772 years (ESPEN), with an individual median follow-up time of ~1 year. The majority of the prospective cohort was female (69.8%), aged < 90 years (80.0%) and had a relatively high BMI (mean 27.0 kg/m^2^, SD:3.9), as residents with a low age-specific BMI were excluded in these analyses. Most frequently described diseases/health-related problems were balance problems (66.7%) and neurological diseases (63.1%). On average, prevalence of diseases was comparable to the cross-sectional cohort. 

Incidence proportions of becoming malnourished during LTCF stay was 13.8% for WL, 11.3% for low BMI, and 8.9% for the ESPEN criteria. The effect sizes of the associations between diseases and/health-related problems and three criteria for malnutrition were relatively comparable (Table 4). Neurological diseases, infectious diseases, balance problems, psychiatric problems, GI tract problems, sleep problems, dental problems, aphasia, and supervised/assisted eating were all associated with (a trend towards) higher incidence of malnutrition, regardless of the malnutrition criterion used. In contrast to the cross-sectional analysis, diabetes showed only a minor lower risk for developing malnutrition based on BMI (HR: 0.73 [95%CI: 0.56–0.93]) and ESPEN (HR: 0.88 [95%CI: 0.63–1.24], and this was not seen for WL (HR: 1.10 [95%CI: 0.90–1.35]. 

Post-hoc analysis showed that incidence of malnutrition increased with a higher number of diseases (HRs ranging from 1.07–1.13).

## 4. Discussion

Residents of long-term care facilities are known to suffer from multiple diseases and health-related problems. In addition, malnutrition is a frequently reported phenomenon. This manuscript indicates that a considerable number of diseases and health-related problems are associated with an increased prevalence, as well as incidence of malnutrition in LTCFs. Herewith, this manuscript is one of the first to describe the associations between diseases and incident malnutrition within LTCFs. 

The characteristics of our study population are relative comparable to other population-representative Dutch cohorts. On average, in the Netherlands, 70% of residents in LTCFs are female, which is equal to the percentage of females in our sample. Residents in our sample were relatively younger (83.1 years) compared to the average Dutch resident in 2019 (85.0 years) [24]. Compared to another Dutch representative cohort [25], our residents had a comparable BMI (24.8 vs. 24.7 kg/m^2^), but the number of diseases was higher in our population (4.5 vs 3.0), which is probably explained by the fact that we also included health-related problems. In general, our population seems to reflect the average population in Dutch LTCFs.

At admission, 850 residents (22.9%) had a low age-specific BMI without recent weight loss (Figure 1). This indicates that older adults have already suffered from an inadequate nutritional status for a longer period. Few specific diseases were related to a low age-specific BMI at admission, indicating that this is a general problem in older adults, regardless of disease status. Screening for early determinants of malnutrition in the community setting and adequate treatment should prevent older adults from entering residential care in a poor nutritional condition [26,27]. 

We showed that the prevalence and (to a lesser extent) incidence of malnutrition are influenced by the used definition; at admission, low age-specific BMI (27.4%) was almost three times higher as weight loss (8.8%). During stay, weight loss (13.8%) was seen more often compared with low age-specific BMI (11.3%) and malnutrition according to ESPEN (8.9%). Associations between diseases/health-related problems and malnutrition also differed between used criteria; WL was more strongly related to diseases than low BMI/ESPEN (except diabetes and cardiac diseases at admission). The use of BMI for diagnosing malnutrition is often debated because of the generally high BMI in Western societies [28]. Our results provide evidence to keep using BMI as an indicator of a poor nutritional status over a longer period. However, to assess incident malnutrition during stays in LTCFs, it is advised to use weight loss, as this better reflects acute nutritional problems. 

Based on the effect sizes of the associations, relatively small differences were seen between the use of BMI alone and the ESPEN criteria for malnutrition. Figure 2 visibly shows that the ESPEN 2015 definition is completely within the spectrum of low age-specific BMI. This dependency on BMI in the ESPEN definition leads to an underestimation of acute nutritional problems. This is clearly illustrated by the large differences in incident malnutrition during stays: 8.9% based on ESPEN vs. 13.8% based on WL. Although WL may indicate an acute problem, dropping below the cut-off points for BMI occurs less frequently, especially in participants with a higher baseline BMI. For example, a resident with a BMI of 27 kg/m^2^ and an average length (1.70 m) should lose ~14.5 kg bodyweight before dropping below the cut-off point of 22 kg/m^2^, which is unlikely to happen, keeping in mind the relatively short stays in LTCFs. Therefore, involuntary WL should always be a trigger to start malnutrition interventions in LTCF residents. 

As also shown in previous research [18], diabetes and cardiac diseases were associated with lower prevalence of malnutrition at admission based on ESPEN and BMI. This can be explained by the high BMIs that usually characterize these diseases. In contrast, diabetes was associated with increased odds on WL, which was also found in another study (OR:1.21 [95%CI: 1.19–1.23]) [29]. However, in both studies, it is unclear whether WL was involuntary or a part of the treatment for diabetes. Our study shows that malnutrition, and especially WL, can also occur in LTCF residents with cardiometabolic diseases, despite a high BMI, and should be monitored and intervened at regular intervals. 

As expected, one of the strongest predictors for developing malnutrition was the need of supervised/assisted eating. Dependency on staff or informal caregivers places residents at risk for a low intake. Assisting residents with eating their meals is time-consuming [30], and shortness of staff, a common problem in residential care facilities, may result in inadequate mealtime assistance [31]. Previously performed trials have shown that increased mealtime assistance resulted in higher intakes [30,32,33]. However, time spent on mealtime assistant in these trials was about 40 min per meal moment, which is far more than in a usual situation. Implementation of these programs would therefore require additional staff and funding. The use of trained volunteers and family members could relieve the pressure on staff [34].

Pressure ulcers, infectious diseases and pain were associated with WL at admission, but to a lesser extent during stay. Treatment of these problems could have decreased the risk of further nutritional decline during stay. In general, the prospective analysis may also have underestimated the malnutrition risk because of misclassification; we only used baseline disease status to study associations with malnutrition, and ignored new diseases that may have occurred during admission. We expect that residents who developed new diseases or health-related problems during their stays were at increased risk of becoming malnourished as well. 

In our analyses, we separately looked at disease groups and at health-related problems. In general, health-related problems (especially psychiatric, balance and sleep problems) were more strongly associated with all three criteria for malnutrition than diseases. We assume that not all diseases were accompanied by acute health problems, especially if residents already suffered from them for dozens of years, Therefore, screening for health-related problems seems to have additional value, next to (long-)diagnosed diseases, when studying the association with malnutrition. 

Our study is one of the first prospective studies that is performed in the LTCF setting. We are aware of a previous study by Torbahn et al. [19], which was based on NutritionDay data in nursing home residents (N = 11,923, follow-up period 6 months), that also tested the association between diseases and malnutrition (BMI < 20 kg/m^2^ and/or recent weight loss (>10% within 6 months)). In this study, comparable effect sizes were seen for immobility (bedbound OR: 1.28 [99.5%CI: 1.00–1.68]) and musculoskeletal diseases (OR: 1.09 [99.5%CI: 0.91–1.31]), but smaller for neurological diseases (OR: 1.10 [99.5%CI: 0.87–1.38]). The difference in effect size for neurological diseases could be explained by the difference in follow-up time (6 months vs. ~1 year in our study), as most neurological diseases have a progressive development [35]. Both our study, as well as the Torbahn et al. [19] study, underline that diseases increase the risk for developing malnutrition during a stay in a LTCF. 

The minimum dataset contains relatively few parameters of malnutrition, as data regarding meal intake, muscle mass, and infection levels are missing. Despite this limitation, routinely collected data (as in InterRAI) can be used to trigger protocols for future assessments. At this moment, the protocol in InterRAI for assessment of malnutrition is only triggered when BMI is below 19. Our data imply that this cut-off point should be raised to 22 and that also those with recent weight loss should be examined for malnutrition by a dietitian. A previous report of our group showed that nearly all residents who displayed recent weight loss or a low BMI also suffered from poor food intake [36]. So, future assessments are strongly recommended, as residents are suffering from recent weight loss or a low BMI.

### Limitations and Strengths

Health-related problems and diseases were analyzed in categories, as larger numbers were needed to investigate single diseases. However, it is not expected that effect sizes will differ strongly between separate diseases in one category, as they are likely to share the same underlying mechanism for their relationship with malnutrition. 

As an outcome of our analysis, we used the ESPEN definition for malnutrition. Recently, the GLIM criteria for malnutrition were launched [22], and it is recommended to use GLIM in future malnutrition studies to create homogeneity in criteria across studies. At this moment, the InterRAI MDS does not provide the necessary items to define malnutrition based on the GLIM criteria; this additionally requires data on reduced muscle mass, reduced food intake or assimilation, and inflammation or disease burden, which are notoriously difficult data to collect in a nursing home population. The use of the ESPEN definition likely underestimated the prevalence and incidence of GLIM malnutrition in our sample, as recent studies showed higher rates of malnutrition based on the GLIM criteria compared to the ESPEN 2015 definition [28,37]. In a recent observational study, we found a malnutrition prevalence rate of 21% in a sample of 176 nursing home residents, based on body weight loss, BMI and registered food intake over 3 days [36], which is indeed higher than ESPEN.

Strong points of our study are the large sample size and prospective design with long follow-up time. In addition, only physician-documented diagnoses are used in InterRAI, making our data more reliable compared to self-reported health status. By reporting associations for both weight loss, low BMI and the ESPEN 2015 criteria, we provide a valuable insight into how diseases affected different criteria for malnutrition and how different criteria for malnutrition affect prevalence rates. Finally, we identified risk factors for developing malnutrition that were not previously described. Our results can contribute to refining malnutrition screening tools, as most risk factors we described are not incorporated in screening tools currently. 

## 5. Conclusions

Most diseases (cardiometabolic diseases excluded) and health-related problems were associated with being malnourished at admission, but also with incidence rates during stay in LTCFs. Moreover, the strength of the associations between diseases or health-related problems and malnutrition are dependent on the set of criteria to define malnutrition. At admission, the use of low age-specific BMI to assess malnutrition status is recommended, as this reflects nutritional status over a longer period. During a LTCF stay, the use of recent weight loss is advised, as this better reflects acute nutritional problems. 

## Figures and Tables

**Figure 1 ijerph-20-03170-f001:**
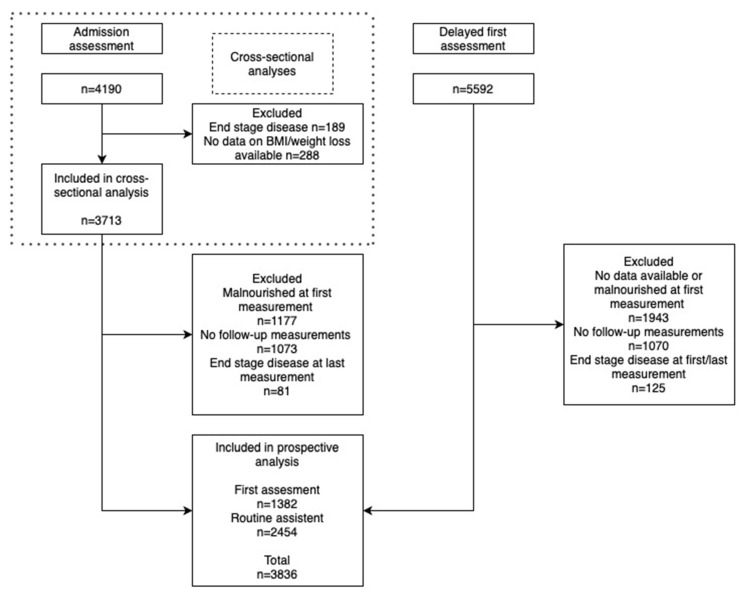
Flow diagram of inclusion and exclusion criteria of residents for cross-sectional and prospective analysis.

**Figure 2 ijerph-20-03170-f002:**
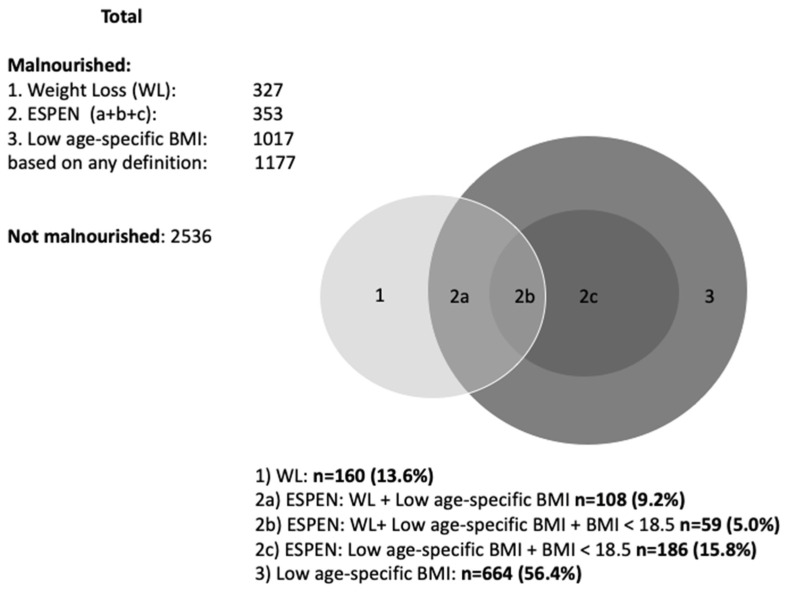
Malnourished residents stratified per definition of malnutrition.

**Table 1 ijerph-20-03170-t001:** Characteristics of included participants for cross-sectional analyses at admission to a LTCF (“newly admitted” residents), stratified by weight loss, low BMI and ESPEN criteria for malnutrition.

	Weight Loss		Low BMI		ESPEN		Total
	Weight Loss Not Present	Weight Loss PRESENT	Normal/High BMI	Low BMI	Well-Nourished	Malnourished	
N (%)	3386 (91.2)	327 (8.8)	2696 (72.6)	1017 (27.4)	3360 (90.5)	353 (9.5)	N = 3713
**Gender**							
Men	1087 (32.1)	119 (36.4)	946 (35.1)	260 (25.6)	1119 (32.3)	87 (24.6)	1206 (32.5)
Women	2299 (67.9)	208 (63.6)	1750 (64.9)	757 (74.4)	2241 (66.8)	266 (75.4)	2507 (67.5)
**Age** (years)							
(mean, SD)	83.1 (SD:7.1)	83.5 (SD:7.0)	82.8 (SD:7.1)	84.1 (SD:6.8)	83.1 (SD:7.1)	83.9 (SD:7.0)	83.1 (SD:7.0)
<90 years	2777 (82.0)	266 (81.3)	2256 (83.7)	787 (77.4)	2772 (82.5)	271 (76.8)	3043 (82.0)
≥90 years	609 (18.0)	61 (18.7)	440 (16.3)	230 (22.6)	588 (17.5)	82 (23.2)	670 (18.0)
**BMI** (mean, SD)	24.9 (SD:4.5)	22.4 (SD:4.7)	26.6 (SD:3.9)	19.7 (SD:1.7)	25.4 (SD:4.3)	18.1 (SD:1.8)	24.7 (SD:4.6)
**Living status before admission**							
Alone	2181 (64.4)	209 (63.9)	1703 (63.2)	687 (67.6)	2155 (64.1)	235 (66.6)	2390 (64.7)
Together	1191 (35.2)	115 (35.2)	980 (36.4)	326 (32.1)	1189 (35.4)	117 (33.1)	1306 (35.3)
**Admission year**							
2005–2009	977 (28.9)	109 (33.3)	764 (28.3)	322 (31.7)	969 (28.8)	117 (33.1)	1086 (29.2)
2010–2014	812 (24.0)	88 (26.9)	675 (25.0)	225 (22.1)	809 (24.1)	91 (25.8)	900 (24.2)
2015–2020	1597 (47.2)	130 (39.8)	1257 (46.6)	470 (46.2)	1582 (47.1)	145 (41.1)	1727 (46.5)
**Diseases**							
**Neurological diseases**	2226 (65.7)	203 (62.1)	1767 (65.5)	662 (65.1)	2213 (65.9)	216 (61.2)	2429 (65.4)
**Musculoskeletal diseases**	242 (7.1)	26 (8.0)	189 (7.0)	938 (92.2)	239 (7.1)	29 (8.2)	268 (7.2)
**Cardiac diseases**	878 (25.9)	99 (30.3)	725 (26.9)	252 (24.8)	900 (26.8)	77 (21.8)	977 (26.3)
**Psychiatric disorders**	626 (18.5)	63 (19.3)	480 (17.8)	209 (20.6)	609 (18.1)	80 (22.7)	689 (18.6)
**Infectious diseases**	450 (13.3)	72 (22.0)	367 (13.6)	155 (15.2)	469 (14.0)	53 (15.0)	522 (14.1)
**Diabetes**	658 (19.4)	73 (22.3)	607 (22.5)	124 (12.2)	693 (20.6)	38 (10.8)	731 (19.7)
**Cancer**	285 (8.4)	51 (15.6)	238 (8.8)	98 (9.6)	297 (8.8)	39 (11.0)	336 (9.0)
**Chronic pulmonary disease**	362 (10.7)	42 (12.8)	290 (10.8)	114 (11.2)	354 (10.5)	50 (14.2)	404 (10.9)
**Pressure ulcers**	256 (7.6)	53 (16.2)	205 (7.6)	104 (10.2)	262 (7.8)	47 (13.3)	309 (8.3)
**Health-related problems**							
**Balance problems**	2360 (69.7)	261 (79.8)	1893 (70.2)	728 (71.6)	2358 (70.2)	263 (74.5)	2621 (70.6)
**Psychiatric problems**	678 (20.0)	70 (21.4)	518 (19.2)	230 (22.6)	661 (19.7)	87 (24.6)	748 (20.1)
**GI tract problems**	885 (26.1)	122 (37.3)	709 (26.3)	298 (29.3)	892 (26.5)	115 (32.6)	1007 (27.1)
**Sleep problems**	1057 (31.2)	127 (38.8)	844 (31.3)	340 (33.4)	1065 (31.7)	119 (33.7)	1184 (31.9)
**Dental problems**	649 (19.2)	92 (28.1)	519 (19.3)	222 (21.8)	650 (19.3)	91 (25.8)	741 (20.0)
**Aspiration**	146 (4.3)	25 (7.6)	115 (4.3)	56 (5.5)	142 (4.2)	29 (8.2)	171 (4.6)
**Fever**	37 (1.1)	8 (2.4)	31 (1.1)	14 (1.4)	40 (1.2)	5 (1.4)	45 (1.2)
**Peripheral edema**	628 (18.5)	63 (19.3)	550 (20.4)	141 (13.9)	645 (19.2)	46 (13.0)	691 (18.6)
**Aphasia**	351 (10.4)	39 (11.9)	290 (10.8)	100 (9.8)	356 (10.6)	34 (9.6)	390 (10.5)
**Pain**	1269 (37.5)	156 (47.7)	1047 (38.8)	378 (37.2)	1291 (38.4)	134 (38.0)	1425 (38.4)
**Locomotion**							
Independently	630 (18.6)	47 (14.4)	466 (17.3)	211 (20.7)	610 (18.2)	67 (19.0)	677 (18.2)
With walking device	1844 (54.5)	153 (46.8)	1487 (55.2)	510 (50.1)	1833 (54.6)	164 (46.5)	1997 (53.8)
Wheelchair	805 (23.8)	93 (28.4)	656 (24.3)	242 (23.8)	803 (23.9)	95 (26.9)	898(24.2)
Bedbound	97 (2.9)	33 (10.1)	80 (3.0)	50 (4.9)	103 (3.1)	27 (7.6)	130 (3.5)
**Supervised/assisted eating**	749 (22.1)	113 (34.6)	583 (21.6)	279 (27.4)	749 (22.3)	113 (32.0)	862 (23.2)
**Number of diseases**	4.4 (SD:2.3)	5.4 (SD:2.5)	4.4 (SD: 2.3)	4.5 (SD:2.3)	4.4 (SD: 2.3)	4.7 (SD: 2.4)	4.6 (SD:2.3)

All characteristics are presented as number with percentage, except age (years), BMI (kg/m^2^) and number of diseases, which are presented as mean with standard deviation. Weight loss: a loss of 5% body mass in the last 30 days or 10% in the last 180 days. Low BMI: <20 kg/m^2^ for residents younger than 70 years and <22 kg/m^2^ for residents 70 years or older. ESPEN: very low BMI (<18.5 kg/m^2^) or weight loss combined with low age-specific BMI as described hereabove.

**Table 2 ijerph-20-03170-t002:** Crude and adjusted * odds ratios for diseases/health-related problems and malnutrition, based on weight loss, low BMI and ESPEN criteria for malnutrition on admission to LTCF (“newly admitted” residents).

	Weight Loss		Low BMI		ESPEN	
	OR	Adjusted OR *	OR	Adjusted OR *	OR	Adjusted OR *
Diseases						
**Neurological diseases**	0.85 [0.67–1.08]	0.89 [0.70–1.14]	0.98 [0.85–1.14]	1.03 [0.88–1.21]	0.82 [0.65–1.02]	0.89 [0.70–1.12]
**Musculoskeletal diseases**	1.12 [0.74–1.71]	1.04 [0.67–1.61]	1.11 [0.85–1.46]	1.04 [0.78–1.38]	1.17 [0.78–1.74]	1.03 [0.68–1.56]
**Cardiac diseases**	1.24 [0.97–1.59]	1.23 [0.96–1.58]	0.89 [0.75–1.05]	0.88 [0.75–1.04]	0.76 [0.58–0.98]	0.76 [0.58–0.99]
**Psychiatric disorders**	1.06 [0.79–1.41]	1.08 [0.81–1.45]	1.19 [0.99–1.43]	1.17 [0.97–1.40]	1.32 [1.01–1.72]	1.29 [0.99–1.69]
**Infectious diseases**	1.84 [1.39–2.43]	1.82 [1.37–2.42]	1.13 [0.92–1.39]	1.10 [0.90–1.36]	1.08 [0.80–1.48]	1.04 [0.76–1.42]
**Diabetes**	1.19 [0.91–1.57]	1.19 [0.90–1.57]	0.48 [0.39–0.59]	0.50 [0.41–0.62]	0.46 [0.33-.066]	0.49 [0.35–0.69]
**Cancer**	2.01 [1.45–2.77]	1.99 [1.44–2.75]	1.10 [0.86–1.41]	1.14 [0.88–1.46]	1.27 [0.89–1.81]	1.32 [0.92–1.88]
**Chronic pulmonary disease**	1.23 [0.87–1.73]	1.19 [0.84–1.69]	1.04 [0.83–1.31]	1.10 [0.88–1.39]	1.40 [1.02–1.93]	1.47 [1.07–2.03]
**Pressure ulcers**	2.36 [1.71–3.25]	2.33 [1.68–3.22]	1.37 [1.07–1.76]	1.36 [1.06–1.75]	1.81 [1.30–2.52]	1.81 [1.30–2.53]
**Health-related problems**						
**Balance problems**	1.72 [1.30–2.27]	1.77 [1.33–2.34]	1.07 [0.91–1.25]	1.05 [0.89–1.23]	1.24 [0.97–1.60]	1.23 [0.96–1.59]
**Psychiatric problems**	1.09 [0.83–1.44]	1.12 [0.85–1.48]	1.22 [1.02–1.45]	1.26 [1.05–1.51]	1.33 [1.03–1.72]	1.40 [1.08–1.81]
**GI tract problems**	1.68 [1.33–2.13]	1.70 [1.33–2.15]	1.15 [0.98–1.35]	1.14 [0.97–1.34]	1.33 [1.05–1.68]	1.32 [1.04–1.68]
**Sleep problems**	1.40 [1.10–1.76]	1.46 [1.15–1.86]	1.10 [0.94–1.28]	1.13 [0.97–1.32]	1.08 [0.86–1.37]	1.15 [0.91–1.46]
**Dental problems**	1.65 [1.28–2.13]	1.73 [1.33–2.24]	1.17 [0.98–1.40]	1.17 [0.97–1.40]	1.44 [1.12–1.85]	1.52 [1.17–1.96]
**Aspiration**	1.83 [1.18–2.85]	2.14 [1.36–3.36]	1.31 [0.95–1.82]	1.40 [1.00–1.96]	2.00 [1.32–3.02]	2.39 [1.56–3.67]
**Fever**	2.27 [1.05–4.91]	2.22 [1.02–4.83]	1.20 [0.63–2.26]	1.36 [0.72–2.58]	1.19 [0.47–3.03]	1.34 [0.52–3.43]
**Peripheral edema**	1.05 [0.79–1.40]	1.11 [0.83–1.48]	0.64 [0.52–0.78]	0.59 [0.48–0.72]	0.64 [0.47–0.89]	0.62 [0.45–0.85]
**Aphasia**	1.17 [0.82–1.66]	1.19 [0.84–1.71]	0.91 [0.72–1.16]	0.97 [0.76–1.24]	0.90 [0.62–1.30]	0.96 [0.66–1.39]
**Pain**	1.52 [1.21–1.91]	1.62 [1.28–2.04]	0.93 [0.80–1.08]	0.91 [0.78–1.05]	0.98 [0.78–1.22]	0.97 [0.77–1.22]
**Locomotion**						
Independently	ref.	ref.	ref.	ref.	ref.	ref.
With walking device	1.11 [0.79–1.56]	1.09 [0.77–1.54]	0.76 [0.63–0.92]	0.65 [0.53–0.79]	0.82 [0.61–1.10]	0.71 [0.52–0.97]
Wheelchair	1.55 [1.07–2.23]	1.45 [1.00–2.11]	0.81 [0.65–1.01]	0.72 [0.58–0.91]	1.08 [0.78–1.50]	0.95 [0.68–1.33]
Bedbound	4.56 [2.78–7.47]	4.80 [2.90–7.96]	1.37 [0.93–2.03]	1.16 [0.78–1.73]	2.39 [1.46–3.91]	2.10 [1.26–3.48]
**Supervised/assisted eating**	1.86 [1.46–2.37]	1.87 [1.46–2.39]	1.36 [1.15–1.60]	1.39 [1.17–1.64]	1.63 [1.29–2.07]	1.63 [1.28–2.07]
**Number of diseases δ**	1.19 [1.14–1.25]	1.21 [1.15–1.27]	1.01 [0.98–1.04]	1.02 [0.98–1.05]	1.05 [1.01–1.10]	1.06 [1.01–1.11]

Data are shown as odds ratios with 95% confidence interval. Weight loss: a loss of 5% body mass in the last 30 days or 10% in the last 180 days. Low BMI: <20 kg/m^2^ for residents younger than 70 years and <22 kg/m^2^ for residents 70 years or older. ESPEN: very low BMI (<18.5 kg/m^2^) or weight loss combined with low age-specific BMI as described hereabove. * adjusted for age category (≤89 years vs. ≥90 years), gender, living status before admission (alone vs. together), year of admission (2005–2009, 2010–2014, 2015–2020). δ used as continues variable with 0 diseases as reference category.

**Table 3 ijerph-20-03170-t003:** Characteristics of included participants for prospective analysis stratified by weight loss, low BMI, and ESPEN criteria for malnutrition.

	Weight Loss Total Follow-Up: 5599 Years Median Follow-Up: 366 Days		Low BMI Total Follow-Up: 5522 Years Median Follow-Up: 360 Days		ESPEN Total Follow-Up: 5772 Years Median Follow-Up: 372 Days		Total
	Did Not Lose Weight	Lost Weight	Normal/High BMI	Low BMI	Stayed Well-Nourished	Became Malnourished	
N (%)	3307 (86.2)	529 (13.8)	3401 (86.7)	435 (11.3)	3626 (91.1)	210 (8.9)	3836
**Gender**							
Men	997 (30.1)	195 (30.6)	1044 (30.7)	115 (26.4)	1101 (30.4)	58 (27.6)	1159 (30.2)
Women	2310 (69.9)	514 (69.4)	2357 (69.3)	320 (73.6)	2525 (69.6)	152 (72.4)	2677 (69.8)
**Age** (years)							
Mean (SD)	83.6 (SD:7.0)	83.9 (SD:6.8)	83.5 (SD:7.0)	84.5 (SD:6.9)	83.6 (SD:7.0)	85.0 (SD:6.6)	83.6 (SD:7.0)
<90 years	2634 (79.6)	560 (81.9)	2741 (80.6)	326 (74.9)	2913 (80.3)	154 (73.3)	3067 (80.0)
≥90 years	673 (20.4)	149 (18.1)	660 (19.4)	109 (25.1)	713 (19.7)	56 (26.7)	769 (20.0)
**BMI** Mean (SD)	27.1 (SD:4.0)	26.7 (SD:3.6)	27.4 (SD:4.0)	24.0 (SD:1.9)	27.2 (SD:4.0)	24.3 (SD:1.7)	27.0 (SD:3.9)
**Living status before admission**							
Alone	2145 (64.9)	349 (66.0)	2205 (64.8)	289 (66.4)	2354 (64.9)	140 (66.7)	2494 (65.0)
Together	1151 (34.8)	180 (34.0)	1185 (34.8)	146 (33.6)	1261 (34.8)	70 (33.3)	1331 (34.7)
missing	11 (0.3)	0	11 (0.3)	0	11 (0.3)	0	11 (0.3)
**Admission year**							
2005–2009	1606 (48.6)	362 (68.4)	1669 (49.1)	299 (68.7)	1808 (49.9)	160 (76.2)	1968 (51.3)
2010–2014	792 (23.9)	96 (18.1)	814 (23.9)	74 (17.0)	858 (23.7)	30 (14.3)	888 (23.1)
2015–2020	909 (27.5)	71 (13.4)	918 (27.0)	62 (14.3)	960 (26.5)	20 (9.5)	980 (25.5)
**Diseases**							
**Neurological diseases**	2063 (62.4)	356 (67.3)	2128 (62.6)	291 (66.9)	2273 (62.7)	146 (69.5)	2419 (63.1)
**Musculoskeletal diseases**	168 (5.1)	18 (3.4)	170 (5.0)	16 (3.7)	182 (5.0)	4 (1.9)	186 (4.8)
**Cardiac diseases**	980 (29.6)	147 (27.8)	1013 (29.8)	114 (26.2)	1069 (29.5)	58 (27.6)	1127 (29.4)
**Psychiatric disorders**	780 (23.6)	129 (24.4)	811 (23.8)	98 (22.5)	867 (23.9)	42 (20.0)	909 (23.7)
**Infectious diseases**	365 (11.0)	73 (13.8)	381 (11.2)	57 (13.1)	407 (11.2)	41 (19.5)	438 (11.4)
**Diabetes**	716 (21.7)	127 (24.0)	769 (22.6)	74 (17.0)	802 (22.1)	41 (19.5)	843 (22.0)
**Cancer**	234 (7.1)	40 (7.6)	241 (7.1)	33 (7.6)	258 (7.1)	16 (7.6)	274 (7.1)
**Chronic pulmonary disease**	413 (12.5)	51 (9.6)	419 (12.3)	45 (10.3)	449 (12.4)	15 (7.1)	464 (12.1)
**Pressure ulcers**	203 (6.1)	17 (3.2)	196 (5.8)	24 (5.5)	210 (5.8)	10 (4.8)	220 (5.7)
**Health-related problems**							
**Balance problems**	2190 (66.2)	369 (69.8)	2268 (66.7)	291 (66.9)	2410 (66.5)	149 (71.0)	2559 (66.7)
**Psychiatric problems**	743 (22.5)	151 (28.5)	769 (22.6)	125 (28.7)	827 (22.8)	67 (31.9)	894 (23.3)
**GI tract problems**	1004 (30.4)	192 (36.3)	1041 (30.6)	155 (35.6)	1115 (30.8)	81 (38.6)	1196 (31.2)
**Sleep problems**	1027 (31.1)	210 (39.7)	1084 (31.9)	153 (35.2)	1153 (31.8)	84 (40.0)	1237 (32.2)
**Dental problems**	610 (18.4)	96 (18.1)	622 (18.3)	84 (19.3)	662 (18.3)	44 (21.0)	706 (18.4)
**Aspiration**	92 (2.8)	14 (2.6)	92 (2.7)	14 (3.2)	98 (2.7)	8 (3.8)	106 (2.8)
**Fever**	30 (0.9)	11 (2.1)	33 (1.0)	8 (1.8)	37 (1.0)	4 (1.9)	41 (1.1)
**Peripheral edema**	748 (22.6)	140 (26.5)	794 (23.3)	94 (21.6)	838 (23.1)	50 (23.8)	888 (23.1)
**Aphasia**	353 (10.7)	67 (12.7)	371 (10.9)	49 (11.3)	392 (10.8)	28 (13.3)	420 (10.9)
**Pain**	1224 (37.0)	189 (35.7)	1253 (36.8)	160 (36.8)	1336 (36.8)	77 (36.7)	1413 (36.8)
**Locomotion**							
Independently	689 (20.8)	123 (23.4)	698 (20.5)	115 (26.4)	757 (20.9)	56 (26.7)	813 (21.2)
With walking device	1847 (55.9)	304 (57.5)	1918 (56.4)	233 (53.6)	2038 (56.2)	113 (53.8)	2151 (56.1)
Wheelchair	703 (21.4)	98 (18.5)	720 (21.2)	81 (18.6)	762 (21.0)	39 (18.6)	801 (20.9)
Bedbound	64 (1.9)	2 (0.4)	61 (1.8)	5 (1.1)	65 (1.8)	1 (0.5)	66 (1.7)
**Supervised/assisted eating**	657 (19.9)	142 (26.8)	677 (19.9)	122 (28.0)	995 (21.9)	139 (31.4)	799 (20.8)
**Number of diseases**	4.4 (SD: 2.4)	4.8 (SD: 2.5)	4.5 (SD:2.4)	4.6 (SD:2.5)	4.4 (SD: 2.4)	4.9 (SD: 2.6)	4.5 (SD: 2.4)

All characteristics are presented as number with percentage, except age (years), BMI (kg/m^2^) and number of diseases, which are presented as mean with standard deviation. Weight loss: a loss of 5% body mass in the last 30 days or 10% in the last 180 days. Low BMI: <20 kg/m^2^ for residents younger than 70 years and <22 kg/m^2^ for residents 70 years or older. ESPEN: very low BMI (<18.5 kg/m^2^) or weight loss combined with low age-specific BMI as described hereabove.

**Table 4 ijerph-20-03170-t004:** Crude and adjusted* hazard ratios (HR) for diseases/health-related problems and malnutrition based on weight loss, low BMI, and ESPEN criteria for malnutrition.

	Weight Loss		Low BMI		ESPEN	
	HR	Adjusted HR *	HR	Adjusted HR *	HR	Adjusted HR *
Diseases						
**Neurological diseases**	1.50 [1.24–1.80]	1.55 (1.29–1.87)	1.43 [1.17–1.75]	1.55 [1.27–1.91]	1.68 [1.25–2.26]	1.89 (1.40–2.54)
**Musculoskeletal diseases**	0.86 [0.54–1.37]	0.86 (0.54–1.37)	0.98 [0.59–1.61]	0.98 [0.59–1.61]	0.49 [0.18–1.32]	0.47 (0.18–1.28)
**Cardiac diseases**	1.04 [0.86–1.26]	1.06 (0.88–1.28)	0.95 [0.77–1.18]	0.96 [0.77–1.19]	1.04 [0.77–1.41]	1.04 (0.77–1.41)
**Psychiatric disorders**	1.06 [0.87–1.29]	1.06 (0.87–1.30)	0.94 [0.75–1.18]	0.94 [0.75–1.17]	0.80 [0.57–1.13]	0.81 (0.58–1.14)
**Infectious diseases**	1.34 [1.04–1.71]	1.30 (1.05–1.61)	1.27 [0.96–1.68]	1.23 [0.93–1.63]	1.46 [1.00–2.14]	1.41 (0.96–2.07)
**Diabetes**	1.09 [0.89–1.33]	1.10 (0.90–1.35)	0.69 [0.53–0.88]	0.72 [0.56–0.92]	0.83 [0.59–1.17]	0.88 (0.63–1.24)
**Cancer**	1.15 [0.84–1.59]	1.13 (0.82–1.56)	1.17 [0.82–1.66]	1.12 [0.79–1.60]	1.18 [0.71–1.97]	1.11 (0.67–1.86)
**Chronic pulmonary disease**	0.76 [0.56–1.01]	0.75 (0.56–1.00)	0.85 [0.62–1.15]	0.85 [0.63–1.16]	0.56 [0.33–0.94]	0.55 (0.33–0.94)
**Pressure ulcers**	0.69 [0.43–1.12]	0.68 (0.42–1.11)	1.19 [0.79–1.79]	1.15 [0.76–1.74]	1.05 [0.55–1.97]	1.03 (0.55–1.95)
**Health-related problems**						
**Balance problems**	1.41 (1.17–1.69)	1.43 (1.19–1.73)	1.16 [0.95–1.42]	1.14 [0.93–1.40]	1.47 [1.09–1.99]	1.49 (1.11–2.02)
**Psychiatric problems**	1.64 [1.35–1.98]	1.63 (1.35–1.97)	1.56 [1.27–1.93]	1.54 [1.25–1.90]	1.88 [1.41–2.52]	1.87 (1.40–2.51)
**GI tract problems**	1.36 [1.14–1.63]	1.35 (1.13–1.62)	1.29 [1.06–1.58]	1.24 [1.02–1.51]	1.49 [1.13–1.97]	1.42 (1.07–1.88)
**Sleep problems**	1.53 [1.29–1.83]	1.53 (1.29–1.83)	1.20 [0.99–1.47]	1.20 [0.98–1.46]	1.52 [1.15–2.00]	1.53 (1.16–2.02)
**Dental problems**	1.15 (0.92–1.44)	1.14 (0.91–1.42)	1.23 (0.97–1.56)	1.20 (0.94–1.52)	1.38 (0.99–1.92)	1.35 (0.96–1.88)
**Aspiration**	1.09 [0.64–1.85]	1.13 (0.66–1.93)	1.38 [0.81–2.35]	1.45 [0.85–2.48]	1.78 [0.88–3.62]	2.02 (0.99–4.13)
**Fever**	2.28 [1.25–4.14]	2.15 (1.18–3.92)	1.84 [0.92–3.71]	1.80 [0.90–3.64]	2.01 [0.75–5.41]	1.94 (0.72–5.23)
**Peripheral edema**	1.18 [0.98–1.44]	1.18 (0.97–1.43)	0.90 [0.71–1.13]	0.87 [0.69–1.10]	1.04 [0.76–1.43]	1.01 (0.73–1.39)
**Aphasia**	1.31 [1.02–1.70]	1.31 (1.01–1.70)	1.11 [0.83–1.50]	1.16 [0.86–1.57]	1.39 [0.93–2.07]	1.50 (1.00–2.25)
**Pain**	1.08 (0.90–1.29)	1.10 (0.92–1.31)	1.09 (0.90–1.32)	1.09 (0.89–1.33)	1.11 (0.84–1.47)	1.13 (0.85–1.51)
**Locomotion**						
Independently	Ref.	Ref.	Ref.	Ref.	Ref.	Ref.
With walking device	0.99	1.01	0.81	0.74	0.81	0.74
	(0.81–1.22)	(0.81–1.25)	(0.65–1.01)	(0.58–0.93)	(0.59–1.12)	(0.53–1.03)
Wheelchair	1.06	1.06	0.92	0.85	0.95	0.88
	(0.82–1.39)	(0.81–1.39)	(0.69–1.22)	(0.63–1.13)	(0.63–1.43)	(0.58–1.33)
Bedbound	0.45	0.45	1.09	1.03	0.52	0.53
	(0.11–1.80)	(0.11–1.84)	(0.44–2.67)	(0.42–2.54)	(0.07–3.79)	(0.07–3.84)
**Supervised/assisted eating**	1.99 (1.64–2.42)	1.96 (1.61–2.39)	1.98 (1.60–2.44)	1.91 (1.55–2.37)	2.45 (1.82–3.29)	2.37 (1.76–3.19)
**Number of diseases δ**	1.12 [1.09–1.16]	1.13 [1.09–1.17]	1.07 [1.03–1.12]	1.07 [1.03–1.11]	1.13 [1.07–1.19]	1.13 [1.07–1.19]

Data are shown as hazard ratios with 95% confidence interval. Weight loss: a loss of 5% body mass in the last 30 days or 10% in the last 180 days. Low BMI: <20 kg/m^2^ for residents younger than 70 years and <22 kg/m^2^ for residents 70 years or older. ESPEN: very low BMI (<18.5 kg/m^2^) or weight loss combined with low age-specific BMI as described hereabove. Adjusted for age category (≤89 years vs. ≥90 years), gender, living status before admission (alone vs. together), year of admission (2005–2009, 2010–2014, 2015–2020). δ used as continuous variable, with 0 diseases as reference category.

## Data Availability

The datasets generated and/or analysed during the current study are not publicly available due to restrictions of the data owner (InterRAI) but are available from the corresponding author on reasonable request.

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
