# Peer review of "Diseases, Health-Related Problems, and the Incidence of Malnutrition in Long-Term Care Facilities"

_ijerph, 2023, doi:10.3390/ijerph20043170_

Round 1
Reviewer 1 Report
I made some comments in a Word document- see attached paper. Thanks!

Author Response
Thank you for your comments. We changed the lines as suggested by you. You can find the changes in the new manuscript.
Reviewer 2 Report
The manuscript has the potential for publication, but it is important to make the corrections suggested below to make it more conceivable.
Abstract:
Line 10: It seems too general to say disease; I suggest saying "certain diseases" instead.
The conclusion in the abstract should be more nuanced and more precisely indicate the significance of the study. The authors should try to emphasize their contribution in recognizing a more reliable method of assessing malnutrition in the elderly, where the risk of malnutrition increases with the appearance of multiple comorbidities.
Introduction
Please define malnutrition and explain its subtypes. Please specify which subtype of malnutrition the authors have focused on in this study.
Methods
Were direct measurements of height and weight performed using anthropometric instruments, or was BMI calculated based on patients' self-reported data about their size and weight?
How are height measurements of immobile patients performed? Were the anthropometric measurements performed according to the protocol (in the early morning hours, after urination and defecation, if possible)?
Results
Line 182. “As shown in table 1, most residents were female (67.6%) …” doesn’t exactly match with the percent shown in table 1 (67,5%). Please check the other numbers.
In the broader context of this definition, someone can be at risk of malnutrition even if he is overweight. That is why it is necessary to specify the subtype of malnutrition. After that, mentioning and excluding cardiovascular diseases from the context of malnutrition, as the authors did in conclusion, will not be necessary.
The study can make a particular contribution to the monitoring of malnutrition in patients during their stay in long-term care facilities. However, it is essential to point out the necessity of dietary tests and biochemical control of nutritional deficits upon admission and during their stay in these institutions.
Author Response
Thank you for your comments. We have made a point-to-point reply and added this in the attachment.

Round 2
Reviewer 2 Report
The authors answered all the questions adequately and considered all remarks where possible. Kudos for the effort.
I would only ask for one explanation:
Are you referring to the references from the old/first version of the manuscript since I don't see any references in the text of the new version manuscript? If that is so, at the publication of Bauer J. et al., 2013 (reference number 6 in the first version of the manuscript), I can't find the part with the explanation you refer to in Lines 42,43,44 of the second version of the manuscript. Please guide me on where exactly to look that.
Anyway, the sentence "Malnutrition may co-occur with obesity; in that case BMIs dropping below cut-off points for malnutrition is less frequently observed(6)" (Lines 42,43,44) is confusing. Please, consider eliminating it.
The same question is for other references as well. Do they change in the new version, or are they the same as in the old one?
Author Response
Thank you again for your time and effort to review our manuscript.
Our reference manager program was not compatible with the reviewed manuscript so we unfortunately did not provide an update reference list in that revised version of our manuscript. Our apologies for that.
In our new version, a reference list is present. So, references are updated compared to the first version of the manuscript and the new references we used for the change we made are now included.
----
Regarding your comment on line 42-44, we changed this towards:
Even when BMI does not drop below clinically relevant cut-off values, for instance in those with obesity(6), involuntary weight loss in older adults is always a risk(7). We hope we addressed your comments